

# SEU²-Net: multi-scale U²-Net with SE attention mechanism for liver occupying lesion CT image segmentation

Lizhuang Liu[1], Kun Wu[1], Ke Wang[2], Zhenqi Han[1], Jianxing Qiu[2], Qiao Zhan[3], Tian Wu[4], Jinghang Xu[4] and Zheng Zeng[4]

[1] Shanghai Advanced Research Institute, Chinese Academy of Sciences, University of Chinese Academy of Sciences, Shanghai, China
[2] Radiology Department, Peking University First Hospital, Beijing, China
[3] Department of Infectious Diseases, The First Affiliated Hospital of Nanjing Medical University, Nanjing, China
[4] Department of Infectious Diseases, Peking University First Hospital, Beijing, China

## ABSTRACT

Liver occupying lesions can profoundly impact an individual's health and well-being. To assist physicians in the diagnosis and treatment of abnormal areas in the liver, we propose a novel network named SEU²-Net by introducing the channel attention mechanism into U²-Net for accurate and automatic liver occupying lesion segmentation. We design the Residual U-block with Squeeze-and-Excitation (SE-RSU), which is to add the Squeeze-and-Excitation (SE) attention mechanism at the residual connections of the Residual U-blocks (RSU, the component unit of U²-Net). SEU²-Net not only retains the advantages of U²-Net in capturing contextual information at multiple scales, but can also adaptively recalibrate channel feature responses to emphasize useful feature information according to the channel attention mechanism. In addition, we present a new abdominal CT dataset for liver occupying lesion segmentation from Peking University First Hospital's clinical data (PUFH dataset). We evaluate the proposed method and compare it with eight deep learning networks on the PUFH and the Liver Tumor Segmentation Challenge (LiTS) datasets. The experimental results show that SEU²-Net has state-of-the-art performance and good robustness in liver occupying lesions segmentation.

## INTRODUCTION

Liver occupying lesion segmentation is a significant research focus in the field of medical image analysis (*Xue et al., 2021*). It not only assists doctors in achieving more accurate analysis, diagnosis, and treatment for liver tumors (*Li et al., 2018*; *Peng et al., 2022*) but also fosters scientific research and technological innovation in the field of medical imaging (*Li & Ma, 2022*). The liver occupying lesion refer to various abnormal structures, tissues, or diseases that appear within the liver tissues, including liver tumors, liver cysts, liver abscesses, and so on. Most research on liver occupying lesion segmentation focus primarily on liver tumor segmentation. The process of liver tumor segmentation is divided into three methods based on gray scale segmentation algorithms, machine learning, and deep

Corresponding authors
Zhenqi Han, hanzq@sari.ac.cn
Zheng Zeng, zeng@bjmu.edu.cn

learning. Based on gray scale segmentation algorithm, *Qi et al. (2008)* proposed a semi-automatic segmentation method of CT liver tumor based on Bayesian rule 3D seed region growing (SRG). In the iterative updating process of region growing, Bayesian decision rules and model matching metrics are used as the growth criteria. This method had a good segmentation effect when the intensity difference between the tumor and normal tissue is large, however, errors are easily introduced when the difference between the tissues is less obvious. *Wong et al. (2008)* proposed a 2D region growing semi-automatic liver tumor segmentation method based on knowledge constraints. The segmentation effect was poor when the occupying lesion tissue and normal liver tissue had low contrast.

For segmentation algorithm based on machine learning, *Das & Sabut (2016)* used the adaptive threshold, morphological processing, and kernel fuzzy C-mean (KFCM) clustering algorithm together with spatial information to segment liver tumors. The segmented liver tumor image of this method had high peak signal-to-noise ratio and low uniform error value. *Rela, Nagaraja & Ramana (2020)* proposed superpixel-based fast fuzzy C-means clustering algorithm (SFFCM) for liver tumor image segmentation to achieve an accuracy of 99.5%. *Anter, Bhattacharyya & Zhang (2020)* presented an optimization method named CALOFCM, which is built upon Fast FCM, chaos theory, and the bio-inspired Ant Lion Optimizer (ALO), to achieve automated localization and segmentation of the liver and liver lesions in CT scans. *Anter & Hassenian (2019)* presented an enhanced segmentation approach for CT liver tumor segmentation, utilizing the watershed algorithm, neutrosophic sets (NS), and the fast fuzzy c-mean clustering algorithm (FFCM). The segmentation algorithms based on machine learning can only achieve good segmentation results on a single dataset but have no transfer ability.

The method based on deep learning realizes automatic liver tumor segmentation, which overcomes the shortcomings of poor segmentation results when the contrast between tumor and normal liver tissue is low in traditional segmentation algorithms. The method based on deep learning has the transfer learning capabilities. Dataset creation is difficult and there are relatively few datasets available for liver tumor image segmentation based on deep learning. However, the proposal of U-Net (*Ronneberger, Fischer & Brox, 2015*) solves the problem of fast training in a small number of data sets. Many improved models based on U-Net have achieved good results in liver tumor segmentation. *Liu et al. (2019)* further improved U-Net by increasing the depth of U-Net and only copying pooling layer features during the skip-connection. Graph segmentation was then used to optimize the segmentation results. This method smoothed the upper boundary of liver tumor segmentation. *Xu et al. (2020)* improved UNet++ and added the residual structure in convolution blocks to avoid the problem of gradient disappearing, which made the Dice coefficient 93.36%. *Seo et al. (2019)* improved the skip connection part of U-Net by adding the residual path with deconvolution layer and activation operation. This method solved the repetition of low-resolution feature information and had a good effect on the segmentation of liver tumor edge and small structure. *Li et al. (2020b)* added an attention mechanism module to the convolution block of UNet++. This method achieved good segmentation results in the LiTS dataset. However, the complex structure led to slow training and prediction speed.

The deep learning methods discussed above have two main problems. some traditional deep learning models such as UNet and SegNet. *Badrinarayanan, Kendall & Cipolla (2017)* only used single-scale feature maps for segmentation, which cannot effectively capture information at different scales. However, when dealing with complex tumor contours and small tumors, the effect is not perfect. Other models often have problems such as boundary blur and excessive smoothing, which make it difficult to accurately segment complex tumor contours and small tumors. Based on these considerations, we choose $U^2$-Net (*Qin et al., 2020*) with multi-scale feature fusion and SE blocks with channel attention mechanism.

We apply $U^2$-Net to the process of liver occupying lesion segmentation, which is a new attempt of $U^2$-Net in the field of medical image processing. In order to better segment small occupying lesion and complex occupying lesion contours, the Squeeze-and-Excitation (SE) module is introduced into the $U^2$-Net architecture (*Li et al., 2020a*; *Gong et al., 2022*). The SE block (*Hu, Shen & Sun, 2018*) is a channel attention mechanism, which can selectively emphasize informative features and suppress less useful ones by explicitly modeling the interdependencies between their convolutional feature channels. Therefore, our model is termed SEU$^2$-Net. We propose an abdominal CT image dataset for liver occupying lesion segmentation from Peking University First Hospital's clinical data (PUFH dataset). The PUFH dataset and Liver Tumor Segmentation Challenge (LiTS) datasets are to train the SEU$^2$-Net model. By testing on the PUFH and LiTS datasets, we prove the superiority of the SEU$^2$-Net model in liver occupying lesion segmentation when compared with different models. SEU$^2$-Net obtains *IoU* 90.55%, *Acc* 99.70%, *Kappa* coefficient 94.89%, *Dice* coefficient 95.04% on the PUFH dataset and *IoU* 83.37 %, *Acc* 99.86%, *Kappa* coefficient 90.86%, *Dice* coefficient 90.93% on the LiTS dataset.

In this study, our primary contributions can be summarized as follows:

(1) We introduce a channel attention mechanism into $U^2$-Net, creating a novel network called SEU$^2$-Net, which aims to achieve precise and automated segmentation of liver occupying lesions.

(2) We propose the Peking University First Hospital (PUFH) dataset, comprising abdominal CT images, which are specifically curated for the segmentation of liver occupying lesions in our study.

(3) Our approach utilizes a hybrid of CrossEntropy and Dice as loss functions, with Dice focusing on enhancing prediction precision and robustness and CrossEntropy contributing to the overall model optimization.

## RELATED WORK

### Multi-scale feature extraction

Multi-scale feature extraction has emerged as a pivotal technique in the field of computer vision and image analysis. This approach recognizes the inherent complexity and diversity of visual information present in images, and addresses it by extracting features at varying scales. The utilization of multi-scale feature extraction enables the model to capture intricate details, from fine-grained textures to larger contextual structures, thus facilitating robust and comprehensive understanding of the input data.

Over the years, a multitude of methods have been proposed to integrate multi-scale information, ranging from pyramidal structures to more advanced deep learning architectures that dynamically learn features at different resolutions. For pyramid in multi-scale feature extraction, *Lin et al. (2017)* introduced the concept of Feature Pyramid Networks (FPN), revolutionizing multi-scale feature extraction in object detection. FPN establishes lateral connections between feature maps from different layers, combining high-resolution details from lower layers with high-level semantics from upper layers. Subsequently, *Zhao & Zhang (2021)* discussed the Enhanced Multi-scale Feature Fusion Pyramid Network for object detection, which emphasizes more effective fusion of multi-scale features, leading to heightened precision in object detection. *Dong, Zhang & Qu (2021)* presented the Multi-path and Multi-scale Feature Pyramid Network (MM-FPN) for object detection. This network likely explores multi-path feature extraction while integrating multi-scale representation within the pyramid structure. Lastly, *Jiao & Qin (2023)* proposed the Adaptively Weighted Balanced Feature Pyramid for object detection. This approach potentially introduces adaptive weighting within the pyramid structure to balance feature contributions and enhance object detection.

U-Net and its variants have achieved tremendous success in biomedical image segmentation, closely tied to their multi-scale feature extraction architecture. The U-Net's encoder–decoder structure allows feature extraction at different scales. The contracting path captures high-level semantics, while the expansive path restores spatial details. This combination excels in delineating objects of varying sizes, making U-Net suitable for tasks like liver lesion segmentation. The Nested U-Net architecture extends U-Net by nesting multiple U-Net blocks within each other. This nested design further enhances multi-scale feature extraction by enabling the network to capture features at various levels of abstraction, improving the model's capability to distinguish objects of diverse sizes and complexities. *Gudhe et al. (2021)* introduces a novel multi-level dilated residual neural network. By incorporating multi-level dilated residual blocks and non-linear residual blocks into skip connections, the network achieves enhanced multi-scale feature extraction capability, leading to improved performance in biomedical image segmentation. Dense U-Net (*Cheng et al., 2021*; *Aldoj et al., 2020*; *Cao et al., 2020*) introduces dense connections in the decoder, linking each layer to all preceding layers, enhancing feature propagation. This architecture aids in capturing multi-scale features and improving the identification of edges and details.

## Attention mechanism

In recent years, attention mechanisms have gained significant prominence across various deep learning tasks, including image segmentation. These mechanisms enhance the network's ability to focus on relevant regions within feature maps, thereby improving both accuracy and interpretability. We introduce self-attention mechanism with both spatial attention and channel attention. The self-attention mechanism can establish relationships between all positions in the input sequence, regardless of a fixed window size and employ multiple heads to simultaneously capture features from different attention perspectives. The Transformer (*Vaswani et al., 2017*) and its variants (*Zhang et al., 2019*; *Mor & Winquist,*

*2002*; *Liu et al., 2021*; *Fu et al., 2019*) utilize the self-attention mechanism, whicn can be modeled for use in sequence-to-sequence tasks. It employs a multi-head self-attention mechanism to capture relationships between different positions within input sequences, thereby enabling modeling of long sequences. The spatial attention mechanism focuses on selectively weighting different spatial positions within an image. By calculating the similarity of each position, it adaptively assigns weights to different spatial locations, enabling the model to emphasize target boundaries and details. In contrast, the channel attention mechanism directs its focus towards different channels within a feature map. By recalibrating the relationships between channels, the channel attention mechanism adaptively emphasizes channels that are more important for the task, while reducing reliance on irrelevant channels. For example, Bottleneck Attention Module (BAM) and Convolutional Block Attention Module (CBAM) incorporate both spatial and channel attention mechanisms. The Squeeze-and-Excitation (SE) block introduces a channel attention mechanism to adaptively recalibrate feature maps.

### Liver occupying lesion CT image segmentation

Numerous techniques have been utilized for liver occupying lesion segmentation (*Anter & Abualigah, 2023*), with a specific emphasis on liver tumor segmentation. As outlined in the Introduction section, the segmentation of liver-occupying lesions can be classified into three main approaches, which rely on grayscale segmentation algorithms, machine learning, and deep learning. Table 1 summarizes the machine learning and deep learning methods applied to liver occupying lesion segmentation.

## MATERIALS AND METHODS

In this section, we design the architecture of $SEU^2$-Net for liver occupying lesion segmentation to test on the PUFH and LiTS dataset. The Biomedical Research Ethics Committee of Peking University First Hospital approved the study of the PUFH dataset (Ethical Review No. 2020 Scientific Research 101 Amendments). The PUFH dataset did not require consent from study participants. Figure 1 shows that the encoder–decoder U-shaped structure of the $SEU^2$-Net consists of 11 SE-RSU structures with different stages, including six encoder stages and five decoder stages. The saliency map fusion module occurs after the decoder levels and the last encoder level. The SE-RSU is still a U-shaped encoder–decoder structure and when compared to the architecture of $U^2$-Net, our SE-RSU adds the channel attention mechanism SE block at the residual connection of RSU. In $SEU^2$-Net, a mix of CrossEntropy and Dice loss functions is used instead of the standard binary cross-entropy loss function used in the $U^2$-Net paper. The hybrid utilization of these two loss functions can enhance the prediction accuracy and model robustness in liver occupying lesion segmentation.

$SEU^2$-Net has attention mechanism and multi-scale feature fusion strategy. For the attention mechanism: $SEU^2$-Net employs an attention mechanism using SE blocks to adaptively weight the channel features, allowing the model to focus on important feature information.The SE block mainly includes the global pooling and sigmod activation functions. Then, the two operations of scale and residual are used to emphasize the

**Table 1   Summary table of liver occupying lesion segmentation methods.**

| Author | Method | Result | Dataset | Year |
|---|---|---|---|---|
| *Das & Sabut (2016)* | adaptive threshold, morphological processing and kernel fuzzy C-mean (KFCM) clustering algorithm | PSNR = 8.5299 | MICCAI 2008 | 2016 |
| *Rela, Nagaraja & Ramana (2020)* | superpixel-based fast fuzzy C-means clustering algorithm | Dice = 0.9154 | 20 CT images | 2020 |
| *Anter, Bhattacharyya & Zhang (2020)* | an optimization method named CALOFCM based on fast-FCM, chaos theory, and bio-inspired ant lion optimizer | Dice = 0.773 | 27 CT images | 2020 |
| *Anter & Hassenian (2019)* | utilizing the watershed algorithm, neutrosophic sets (NS), and the fast fuzzy c-mean clustering algorithm | Dice = 0.9288 | 30 CT images | 2019 |
| *Liu et al. (2019)* | increasing the depth of U-Net and only copying pooling layer features during skip-connection and use graph segmentation to optimize the results | Dice = 0.9505 | codalab | 2019 |
| *Xu et al. (2020)* | improved UNet++ and added the residual 68 structure in convolution blocks to avoid the problem of gradient disappearing. | Dice = 0.9336 | from 15 patients | 2020 |
| *Seo et al. (2019)* | improved the skip connection part of U-Net by adding 70 the residual path with deconvolution layer and activation operation | Dice = 0.8972 | LiTS | 2019 |
| *Li et al. (2020a)* | added an attention mechanism module to the convolution block 73 of UNet++ | Dice = 0.9815 | LiTS | 2020 |
| *Ahmad et al. (2019a)* | training the deep belief network through unsupervised pretraining and supervised fine-tuning | Dice = 0.9480<br>Dice = 0.9183 | Sliver07 3Dircadb01 | 2019 |
| *Ahmad et al. (2022)* | a very lightweight convolutional neural network and Gaussian distribution for weight initialization | Dice = 0.95<br>Dice = 0.929<br>Dice = 0.9731 | Sliver07 3Dircadb01 LiTS | 2022 |
| *Ahmad et al. (2019b)* | a new approach called CNN-LivSeg | Dice = 0.9541 | Sliver07 | 2019 |

occupying lesion information in liver occupying lesion segmentation, especially the small occupying lesion and complex occupying lesion contour.

For multi-scale feature fusion strategy: Firstly, the SEU$^2$-Net model adopts multi-scale images input in the encoder. Specifically, it scales the original image to different scales by a certain proportion, and inputs the scaled image into the model for feature extraction and segmentation prediction, whose advantage is that the image features at different scales can be effectively extracted, and the cross-layer fusion between the features at different scales can be performed to further improve the accuracy and robustness of the model. SEU$^2$-Net will obtain the feature map of the corresponding scale from the encoder for each decoder layer, and fuse it into the feature map of the current layer through upsampling and deconvolution operations. The feature information of different scales can be used to improve the detection and segmentation ability of the model for objects.

## Residual U-block with squeeze-and-excitation

Residual U-block with Squeeze-and-Excitation (SE-RSU) is a new structure introduced in U$^2$-Net as shown in Fig. 1, which incorporates the SE mechanism into the RSU of U$^2$-Net. The SE-RSU terms that the SE block is added to the position of the residual connection in the RSU. The SE block is an attention mechanism used to enhance the model's feature representation capability by adaptively learning the correlations between feature channels

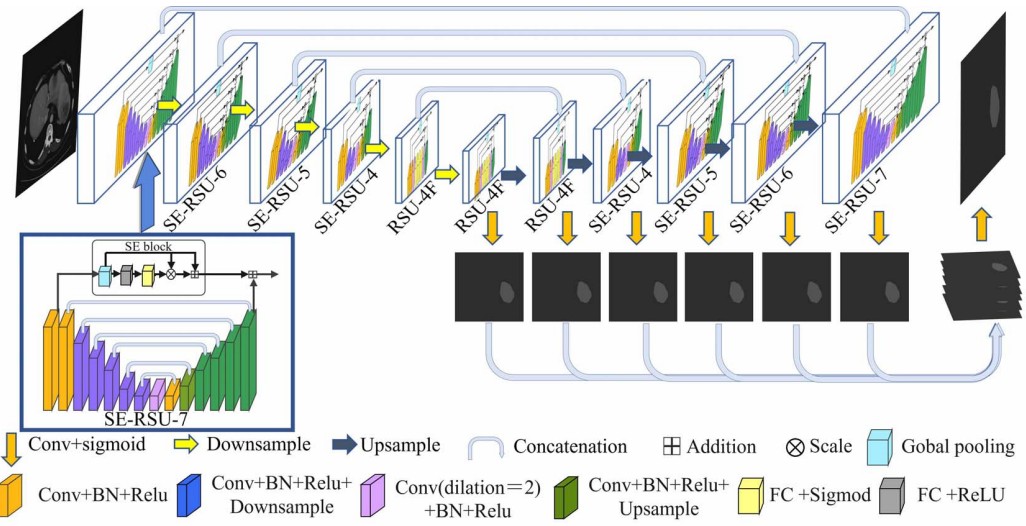

**Figure 1** **The architecture of SEU²-Net.** SEU²-Net features a symmetrical structure comprising eight SE-RSU blocks and three RSU blocks. Each block represents an encoder–decoder structure and is annotated with its respective name. The numerical labels "7", "6", "5", and "4" correspond to the number of layers in the encoder of each block. Progressing from "7" to "4", the encoder undergoes a reduction of one encoding layer, characterized by Conv+BN+Relu+Downsample. The term "SE" signifies the integration of the SE attention mechanism within the block. Additionally, the letter "F" denotes the substitution of pooling and upsampling operations with dilated convolutions, ensuring that all intermediate feature maps of RSU-4F maintain the same resolution as their input feature map. The input image is a 512 × 512 abdominal CT slice, and the output image is a 512 × 512 segmented liver occupying lesion image. (Image source credit: LiTs dataset, CC BY NC-ND 4.0).

to better capture important feature information, as shown in Fig. 2. The purpose of adding SE after the first convolutional block of SE-RSU is coarse-grained context detection in the initial layer network and fine-grained context detection in the deep network as SE-RSU goes deeper in the SEU²-Net network.

Figure 2 shows that the SE block is implemented by passing the output of the convolution operation through a global pooling layer, two fully-connected layers (FC) and and activation function operations in SE-RSU. Firstly, global spatial information compression into channel descriptors is achieved by using global average pooling to generate channel statistics. *i.e.,* the size $H \times W \times C$ feature is compressed to $1 \times 1 \times C$. The role of the fully connected layers is to restrict model complexity, promote generalization, and perform linear transformations to facilitate the activation function in capturing inter-channel correlations more effectively. Then, the ReLU and sigmoid activation function is used to learn nonlinear interactions between channels to capture channel dependencies. The output of sigmod function is channel-multiplied with the original feature to highlight the useful feature information and ignore the less useful ones. Finally, residual connection in the SE block is used to avoid gradient vanishing. Equation (1) shows that $W_1$ and $W_2$ represent the weights of two FC layers, and $ReLU(x)$ is defined as $\max(0,x).\sigma$ represents the sigmoid activation function. The $H$, $W$ and $C$ stand for the height, width, and depth (number of channels) of the feature map respectively, and $X_c(i,j)$ denotes the value of the feature map at position $(i,j)$

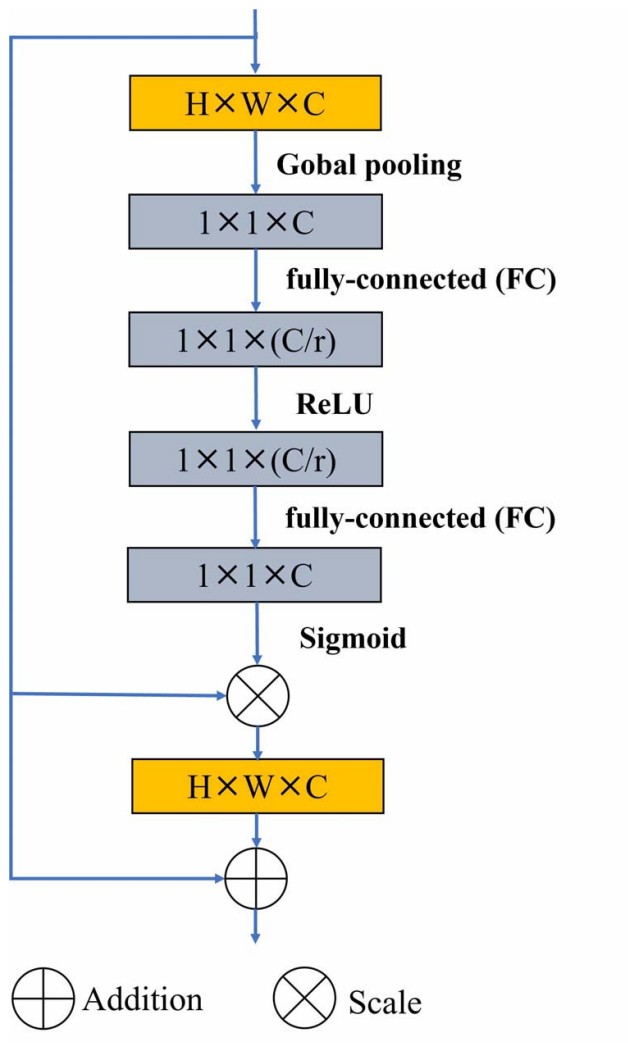

**Figure 2** **The SE block includes global pooling and sigmod activation operations, which is added to the residual connection of the SE-RSU.**

of the $c_{th}$ channel.

$$SE = \sigma \left( W_2 \cdot \max \left( 0, W_1 \cdot \frac{1}{H \times W} \sum_{i=1}^{H} \sum_{j=1}^{W} X_c(i,j) \right) \right) \odot X_c(i,j) + X_c(i,j). \tag{1}$$

## Loss functions

The standard binary cross-entropy loss function is used to describe U²-Net, however, we use a combination of CrossEntropy and Dice Loss functions in our approach. The Dice Loss function is primarily used to optimize the model's prediction accuracy and robustness, particularly for enhancing the prediction precision and robustness of the model. The CrossEntropy Loss function is renowned for its ability to measure pixel-level class probability discrepancies. By using these two loss functions together, the model

can find a balance between accuracy and precision and has better generalization ability. However, the binary cross entropy loss function is sensitive to pixel class distribution imbalance, which can cause the model's prediction results to be biased towards regions with more pixel classes, and can result in misclassification in regions with fewer pixel classes. Additionally, the binary cross entropy loss function cannot handle the problem of target boundaries well, which can lead to prediction results with breaks or blur at the boundaries, affecting the segmentation performance of the model. Therefore, the binary cross entropy loss function is not effective for small or complex occupying lesion contours in liver occupying lesion segmentation.

Choose to use a mix of CrossEntropy and Dice loss functions with corresponding weights of 0.7 and 0.3. Equation (2) is the loss function expression of CrossEntropy Loss. $N$ represents the number of samples, $C$ represents the number of classes, $y_{ij}$ represents the $j$th label of $i$th sample, and $p_{ij}$ represents the predicted probability of the $j$th class for $i$th sample. The cross-entropy loss function measures how close the predicted value is to the ground truth.

$$CrossEntropyLoss = -\frac{1}{N}\sum_{i=1}^{N}\sum_{j=1}^{C} y_{ij} * log(p_{ij}). \tag{2}$$

Equation (3) is the expression of the loss function for Dice Loss, where Dice is given by Eq. (7). The Dice Loss function is used when the background area is much larger than the target area. Because the liver occupying lesion area is smaller than the liver background area, the Dice loss function will have a better effect.

$$DiceLoss = 1 - \frac{|A \cap B|}{|A| + |B|}. \tag{3}$$

## Evaluation metrics

*IoU* refers to the ratio between the intersection and union of the true value and the predicted value in a class. The formula is as shown in Eq. (4). Here, *TP* represents the intersection of the predicted liver occupying lesion and labeled occupying lesion. $TP + FP + FN$ represents the union of the predicted liver occupying lesion and labeled occupying lesion.

$$IoU = \frac{TP}{TP + FP + FN}. \tag{4}$$

Accuracy (*Acc*) is the overall classification accuracy, which is the probability of predicting the correct number of samples over the total number of samples. *Acc* is expressed in Eq. (5) using confusion matrices. The *intersect_area* represents the intersection area of prediction and ground truth on all classes. All classes in the experiment refers to the liver space-occupying lesion and background. *pred_area* represents the prediction area on all classes.

$$Acc = \frac{intersect\_area}{pred\_area}. \tag{5}$$

The *Kappa* coefficient measures whether the predicted value is consistent with the actual classification value. The *Kappa* coefficient compensates for the bias towards large categories

in Acc caused by the gap in the number of categories. Equation (6) is the calculation method of *Kappa* coefficient, where $P_e$ represents the ratio of the product of the number of true categories and the number of corresponding predicted categories to the square of the total number of samples. A Kappa coefficient between 0.81 and 1 indicates almost perfect agreement.

$$Kappa = \frac{Acc - P_e}{1 - P_e}. \tag{6}$$

The *Dice* coefficient represents the proportion of duplicate parts between the predicted segmented image and the annotated image. *Dice* has a value between 0 and 1. In Eq. (7), *A* represents the pixel value of the true image and *B* represents the pixel value of the predicted image.

$$Dice = \frac{|A \cap B|}{|A| + |B|}. \tag{7}$$

## Process of experiment

The datasets were divided into the training set, test set, and verification set in a ratio of 8:1:1, respectively. The Cross Entropy and Dice Loss functions are combined. The momentum optimizer which contains the Newtonian momentum flag can better eliminate the wobble phenomenon in the process of updating the hyperparameter. We use cosine annealing methods to dynamically adjust the learning rate as shown in Eq. (8).

$$\eta_t = \eta^i_{min} + \frac{1}{2}(\eta^i_{max} - \eta^i_{min})(1 + \cos\frac{T_{cur}\pi}{T_i}) \tag{8}$$

Let *i* denote the number of learning rate changes at the $i_t h$ time. The initial learning rate $\eta^i_{max}$ is 0.0015. The minimum learning rate $\eta^i_{min}$ is 0. $T_{cur}$ is the current epoch. $T_i$ is the number of iterations in one learning rate cycle, which is set to 5400. The batch size is 4. The model is saved every 900 training runs, for a total of 9,000 training runs. The experiments are trained on an NVIDIA Tesla V100 GPU (32 GB).

## RESULTS

### PUFH dataset

#### Dataset setup

The PUFH dataset consisted of 200 3D abdominal CT images and corresponding labeled images. The images ared labeled to highlight various types of liver occupying lesions such as hepatic cysts, liver abscess, and hepatocellular carcinoma. The number of slices that can be scanned at a time ranges from 39 to 107 depending on the information of each 3D image when the dataset is sliced horizontally. The resolution of the slice images ranged from (0.5839844, 0.5839844, 5.0) to (0.88671875, 0.88671875, 5.0) mm, and the image intensity ranged from (−1024.0, 3071.0) to (−3024.0, 1210.0). The slice size is 512 × 512. Figure 3 shows the fusion slice image of a random abdominal CT image and its corresponding annotated image in the PUFH dataset using ITK-Snap.

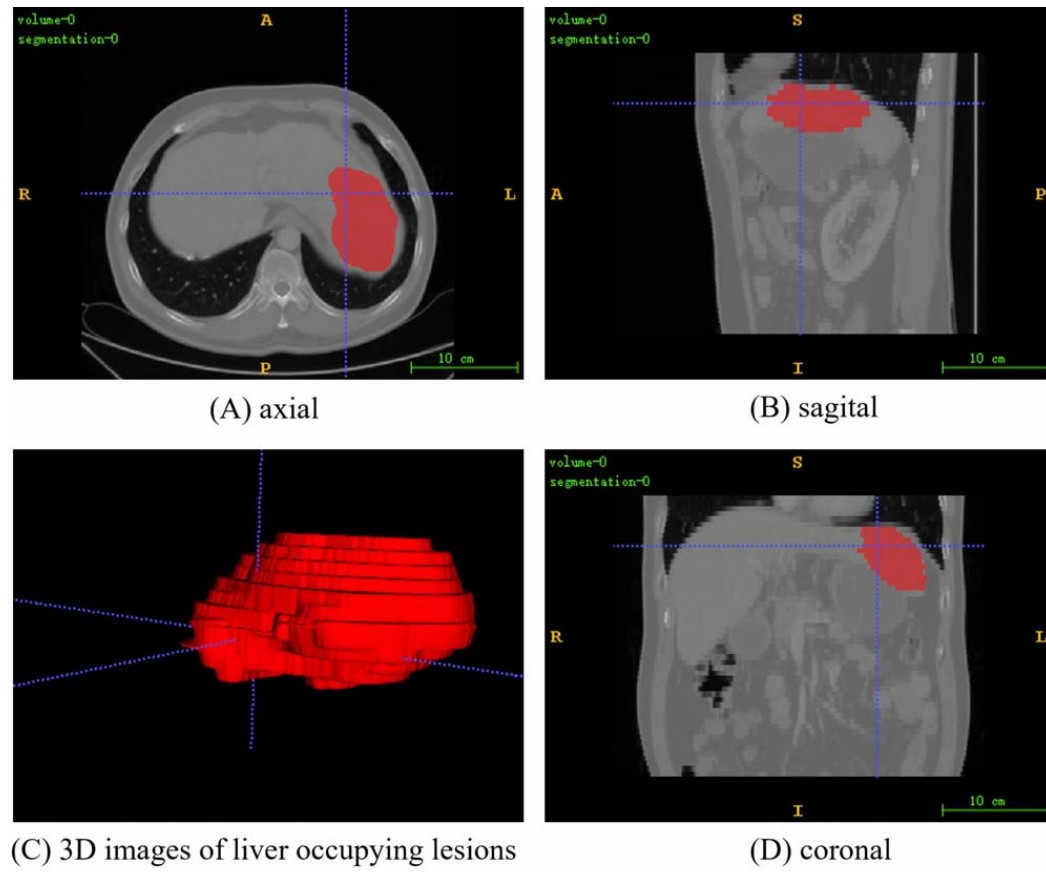

**Figure 3** **A random fusion CT image from PUFH dataset, resulting from the merging of abdominal CT images and their corresponding annotated images.** The fused image comprises a total of 53 slices with a volume voxel size of (0.6836,0.6836,5). (A), (B), and (D) represent the axial, sagittal, and coronal planes of the fused images, respectively. (C) showcases a 3D image depicting the annotated regions of liver-occupying lesions (Image source credit: LiTs dataset, CC BY NC-ND 4.0).

### Dataset processing

The dataset is comprised of abdominal CT images containing multiple organs. In order to ensure the accuracy of the model training, the slice data of the original image from the beginning to the end of the liver is found according to the labeled image data. The Nii format data containing only the liver region is converted into 2D data in PNG format, and the size of the converted 2D data is $512 \times 512$.

### Experimental results

We compared SEU$^2$-Net, an improved model derived from U$^2$-Net and a member of the U-Net family, with U-Net, U$^2$-Net, and its variants, including AttentionU-Net (*Oktay et al., 2018*), UNet3Plus (*Huang et al., 2020*), and UNetPlusPlus (*Zhou et al., 2020*). Additionally, to assess SEU$^2$-Net's superior performance, we compared it with other recent state-of-the-art semantic segmentation techniques, including DDRNet _23 (*Pan et al., 2022*), DNLNet (*Yin et al., 2020*), and GSCNN (*Takikawa et al., 2019*). Table 2 shows the evaluation metrics

**Table 2** Evaluation metrics table of PUFH dataset.

|  | IoU (%) | Acc (%) | Kappa (%) | Dice (%) |
|---|---|---|---|---|
| U-Net | 86.85 | 99.54 | 92.72 | 92.96 |
| AttentionU-Net | 82.06 | 99.36 | 89.81 | 90.14 |
| UNet3Plus | 65.10 | 98.53 | 78.10 | 78.86 |
| UNetPlusPlus | 81.92 | 99.42 | 89.76 | 90.06 |
| $U^2$-Net | 90.11 | 99.68 | 94.63 | 94.80 |
| $SEU^2$-Net (Ours) | **90.55** | **99.70** | **94.89** | **95.04** |
| DDRNet _23 | 83.90 | 99.49 | 90.98 | 91.24 |
| DNLNet | 83.99 | 99.48 | 91.03 | 91.30 |
| GSCNN | 84.22 | 99.49 | 91.17 | 91.43 |

**Notes.**
Bold styling indicates the maximum or highest values.

of PUFH dataset in nine models. $SEU^2$-Net outperforms for liver CT image segmentation in five methods, achieving the *IoU* ratio increased 0.44%, the *Acc* increased 0.02%, the *Kappa* coefficient increased 0.26%, the *Dice* coefficient increased by 0.24% over $U^2$-Net. Compared with AttentionU-Net, which is the combination of attention mechanism and UNet, $SEU^2$-Net had the highest improvement, among which the *IoU* ratio increased by 8.49%, the *Acc* increased by 0.34%, the *Kappa* coefficient increased by 5.08% and the *Dice* coefficient increased by 4.9%. When compared with DDRNet _23, DNLNet, and GSCNN, it is evident that $SEU^2$-Net consistently outperforms these models by a considerable margin.

Figure 4 shows that six 2D slices are randomly selected in the dataset to observe the liver occupying lesion segmentation effect of the nine models. The effect of liver occupying lesion segmentation depends on the location of segmentation, the overall contour and the recognition of small liver occupying lesion. $SEU^2$-Net is most similar to label in terms of the location and size of liver occupying lesion. For example, the liver occupying lesions in the fifth slice has complex contours and fine regions. Comparing the segmentation effects of the six models in the fifth slice, $SEU^2$-Net can identify very small liver occupying lesion and complex edges (the mark of the red box), which greatly illustrates the importance of $SEU^2$-Net's attention mechanism and its ability to capture feature information from multiple scales perspectives.

## LiTS dataset
### Dataset setup and processing
Liver tumors are a type of liver lesion, therefore, the LiTS dataset of 131 contrast-enhanced 3D abdominal CT scans includes 70 scans for training and testing with corresponding annotated images specifically focusing on liver occupying lesion segmentation. The number of tumors varied from 0 to 75, and the size varied from 38 $mm^3$ to 349 $mm^3$. The layer spacing (section thickness) is (0.45,6) mm. The number of slices that can be scanned at a time ranges from 42 to 1,026 according to the different information of each 3D image. The size of 2D slices is $512 \times 512$, where the planar resolution is ($0.6 \times 0.6$, $1.0 \times 1.0$) mm.

For the processing of the LiTS dataset, only the tumor label is retained during liver tumor segmentation because the dataset contained both liver and tumor labels. Then, the

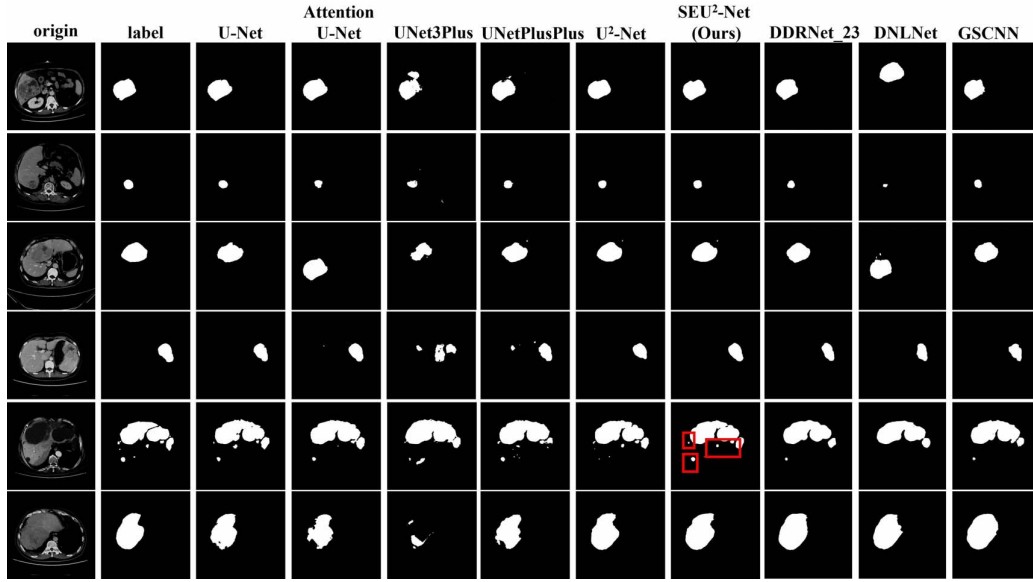

**Figure 4** **Liver occupying lesion segmentation of PUFH dataset.** Column 1: Origin. Column 2: Label. Column 3: U-Net. Column 4: AttentionU-Net. Column 5: UNet3Plus. Column 6: UNetPlusPlus. Column 7: $U^2$-Net. Column 8: $SEU^2$-Net. Column 9: DDRNet _23. Column 10: DNLNet. Column 11: GSCNN (Image source credit: LiTs dataset, CC BY NC-ND 4.0).

processing of the dataset is the same as that of the PUFH dataset. Figure 5 shows the fusion slice image of a random abdominal CT image and its corresponding annotated image in the LiTS dataset.

*Experimental results*

The nine models trained on the PUFH dataset are used for training and testing in the LiTS dataset. The evaluation metrics of LiTS dataset are shown in Table 3. Compared with $U^2$-Net, $SEU^2$-Net has the *IoU* increased by 2.7%, *Kappa* coefficient increased by 1.65%, and *Dice* coefficient increased by 1.63% in the evaluation metrics. UNet3Plus performs the worst among the evaluation metrics. Compared with UNet3Plus, the *IoU* of $SEU^2$-Net is increased by 20.11%, *Acc* is increased by 0.23%, the *Kappa* coefficient is increased by 13.55%, and the *Dice* coefficient is increased by 13.44%. At the same time, the evaluation metrics of $SEU^2$-Net on LITS dataset are still higher than those of AttentionU-Net. When comparing $SEU^2$-Net against non-UNet variants within the LiTS dataset, its outstanding performance is clearly demonstrated. Figure 6 shows six randomly selected 2D slices in the LiTS dataset to observe the liver tumor segmentation effect of the nine models. Most of liver tumors in these six pictures are small, and $SEU^2$-Net is most similar to label in terms of the location and size of liver tumors, which proves that $SEU^2$-Net has advantages in the segmentation of small tumors.

## Model complexity

$SEU^2$-Net exhibits outstanding performance on both datasets in terms of evaluation metrics. However, in order to evaluate the model complexity, we conduct a comprehensive

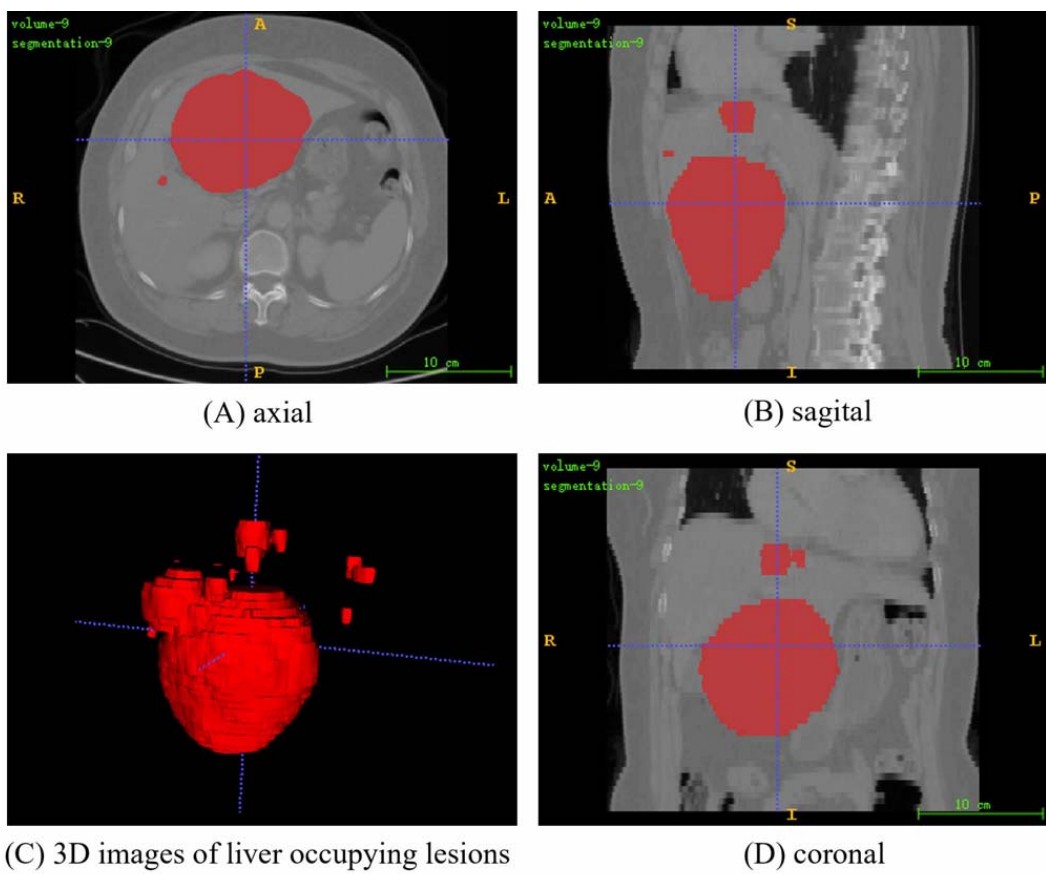

(A) axial

(B) sagital

(C) 3D images of liver occupying lesions

(D) coronal

**Figure 5** **A random fusion CT image from LiTS dataset, resulting from the merging of abdominal CT images and their corresponding annotated images.** The fused image comprises a total of 55 slices with a volume voxel size of (0.584,0.584,5). (A), (B), and (D) represent the axial, sagittal, and coronal planes of the fused images, respectively. (C) showcases a 3D image depicting the annotated regions of liver-occupying lesions. (Image source credit: LiTS - Liver Tumor Segmentation Challenge, CC BY NC-ND 4.0).

analysis of this model. This crucial for gaining a more extensive understanding of the model's characteristics, as it encompasses two key components: computational cost, typically quantified in floating point operations (FLOPs), and the number of model parameters (Params). In this section, we delve into the model complexity of SEU$^2$-Net and compare it with several state-of-the-art semantic segmentation models, including UNet, AttentionU-Net, UNet3Plus, UNetPlusPlus, U$^2$-Net, DDRNet _23, DNLNet, and GSCNN.

Table 4 presents a comparison between SEU$^2$-Net and eight other models in terms of FLOPs and Params for network complexity.The maximum values in FLOPs and Params are highlighted in bold in the table. It becomes evident that SEU2Net demonstrates a comparatively higher level of computational complexity. This is primarily attributed to the inherently greater complexity of the U2Net model itself. When comparing the model complexities of SEU$^2$-Net and U$^2$-Net, it is evident that SEU$^2$-Net's computational complexity is only marginally higher than that of U$^2$-Net. Specifically, SEU$^2$-Net exhibits an

**Table 3** Evaluation metrics table of LiTS dataset.

|  | IoU (%) | Acc (%) | Kappa (%) | Dice (%) |
|---|---|---|---|---|
| U-Net | 75.57 | 99.78 | 85.98 | 86.08 |
| AttentionU-Net | 75.98 | 99.78 | 86.24 | 86.35 |
| UNet3Plus | 63.26 | 99.63 | 77.31 | 77.49 |
| UNetPlusPlus | 73.46 | 99.75 | 84.57 | 84.70 |
| $U^2$-Net | 80.67 | 99.83 | 89.21 | 89.30 |
| $SEU^2$-Net (Ours) | **83.37** | **99.86** | **90.86** | **90.93** |
| DDRNet _23 | 74.87 | 99.78 | 85.52 | 85.63 |
| DNLNet | 64.72 | 99.68 | 78.42 | 78.58 |
| GSCNN | 69.37 | 99.72 | 81.78 | 81.91 |

**Notes.**
Bold styling indicates the maximum or highest values.

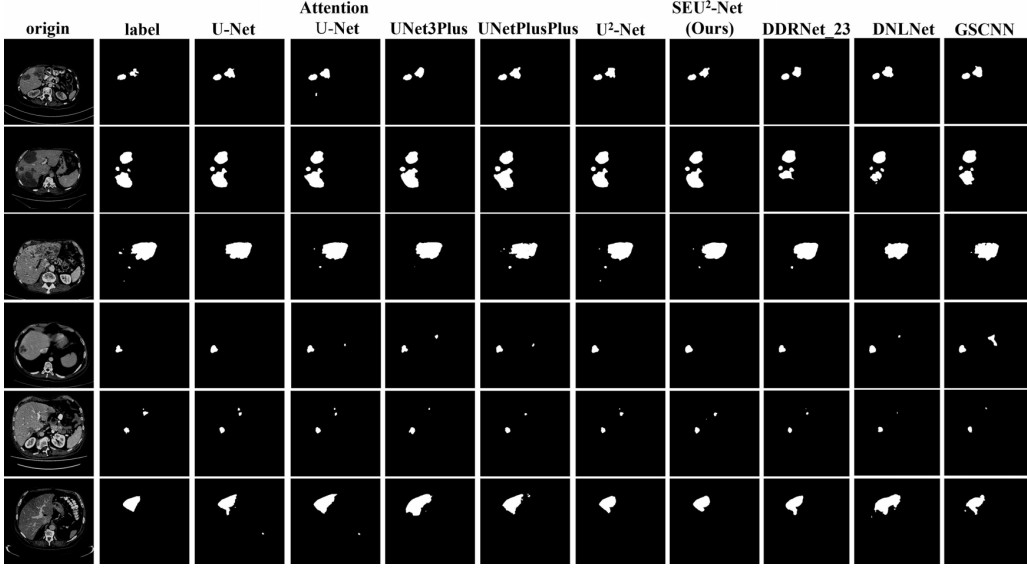

**Figure 6** **Liver occupying lesion segmentation of LiTS dataset.** Column 1: Origin. Column 2: Label. Column 3: U-Net. Column 4: AttentionU-Net. Column 5: UNet3Plus. Column 6: UNetPlusPlus. Column 7: $U^2$-Net. Column 8: $SEU^2$-Net. Column 9: DDRNet _23. Column 10:DNLNet. Column 11: GSCNN. (Image source credit: LiTS - Liver Tumor Segmentation Challenge, CC BY NC-ND 4.0).

increase of 1,152 FLOPs and 1,184 Params compared to $U^2$-Net. This marginal increment in complexity can be attributed to the incorporation of SE blocks into all eight SE-RSU blocks within $SEU^2$-Net, resulting in an additional computational load of 148 FLOPs and 144 Params for each SE block. Despite this slight increase in computational demand, $SEU^2$-Net achieves a noteworthy 1% enhancement in *IoU* on the LiTS dataset compared to $U^2$-Net. This emphasizes the effectiveness of SE blocks in boosting segmentation performance while maintaining the model's relative lightweight characteristics.

**Table 4**  Table of model complexity comparison, Giga FLOPs represents one billion floating-point operations per second.

|  | Giga FLOPs | Params |
| --- | --- | --- |
| U-Net | 124.30 | 13,404,354 |
| AttentionU-Net | 266.29 | 34,894,262 |
| UNet3Plus | **792.22** | 26,987,714 |
| UNetPlusPlus | 119.66 | 8,368,872 |
| $U^2$-Net | 150.69 | 44,052,518 |
| $SEU^2$-Net (Ours) | 150.69 | 44,053,702 |
| DDRNet _23 | 17.93 | 20,184,130 |
| DNLNet | 209.67 | **50,069,093** |
| GSCNN | 192.68 | 39,460,763 |

**Notes.**
Bold styling indicates the maximum or highest values.

**Table 5**  The comparative table between the Mixture, CrossEntropy, Dice, and BCE Loss functions using the PUFH dataset.

|  | IoU (%) | Acc (%) | Kappa (%) | Dice (%) |
| --- | --- | --- | --- | --- |
| CrossEntropy loss | 90.22 | 99.69 | 94.70 | 94.86 |
| Dice loss | 60.25 | 98.59 | 74.48 | 75.2 |
| BCE loss | 90.08 | 99.69 | 94.62 | 94.78 |
| Mixture loss (ours) | **90.55** | **99.70** | **94.89** | **95.04** |

**Notes.**
Bold styling indicates the maximum or highest values.

## Loss function analysis

In the context of liver occupying lesion segmentation, the choice of an appropriate loss function plays a crucial role in determining model performance. We employed a combination of CrossEntropy and Dice Loss functions, denoted as mixture loss. Mixture loss offers a synergistic advantage by promoting prediction precision and robustness in segmentation tasks. To ascertain the superiority of our novel hybrid loss function over each individual loss function, we conducted the loss function ablation experiment. Additionally, we consider binary cross-entropy loss function (BCE Loss) suitable for binary segmentation tasks, since it is adopted by the authors of $U^2$-Net as the loss function. To provide a comprehensive evaluation, we compared against CrossEntropy Loss, Dice Loss, and BCE Loss individually in the PUFH and LiTS dataset. Detailed experimental results can be found in Tables 5 and 6. The highest value in the evaluation metrics is highlighted in bold.

It becomes evident that mixture loss outperforms all other loss functions, including the CrossEntropy, Dice, and BCE Loss functions, across all evaluation metrics (IoU, Acc, Kappa, Dice) for both the PUFH and LiTS datasets. This consistent superiority in segmentation performance on both datasets demonstrates that Mixture Loss excels in accurately capturing the characteristics and boundaries of liver occupying lesions. To visualize the evolution of loss functions during model training, Figs. 7 and 8 illustrate the changing curves of the four loss functions throughout the training process, accompanied by

**Table 6** The comparative table between the Mixture, CrossEntropy, Dice, and BCE Loss functions using the LiTS dataset.

|  | IoU (%) | Acc (%) | Kappa (%) | Dice (%) |
|---|---|---|---|---|
| CrossEntropy loss | 82.22 | 99.84 | 90.16 | 90.24 |
| Dice loss | 36.81 | 99.20 | 53.41 | 53.81 |
| BCE loss | 82.95 | 99.85 | 90.61 | 90.68 |
| Mixture loss (ours) | **83.37** | **99.86** | **90.86** | **90.93** |

**Notes.**
Bold styling indicates the maximum or highest values.

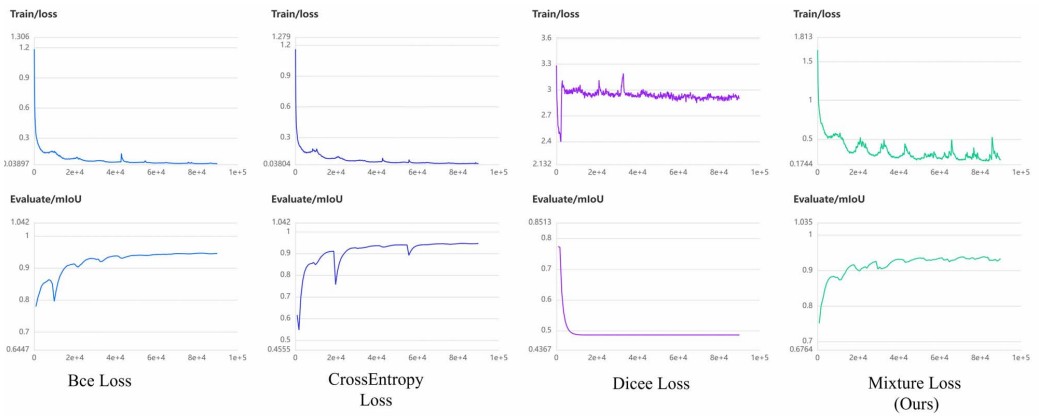

**Figure 7** The loss function and mIoU variation curves for Mixture, CrossEntropy, Dice, and BCE Loss functions on the PUFH dataset.

the corresponding fluctuations in *mIoU* on the validation dataset. Based on the two images, we can observe that when Dice Loss is used independently, there is a phenomenon of loss function non-convergence and a very poor performance in terms of evaluation metrics. The use of CrossEntropy or BCE Loss functions individually leads to fewer oscillations in the curves, it is evident that Mixture Loss outperforms all others in terms of all evaluation metrics.

## DISCUSSION

Automatic segmentation of liver occupying lesion plays an important role in the clinical diagnosis and treatment of liver diseases. The ability to accurately determine the location and size of the lesions greatly improves the efficiency of diagnosis. In this article, we propose the SEU$^2$-Net deep learning model, which adds the SE block with channel attention mechanism to the U$^2$-Net model that can capture context information from multiple scales. In order to verify the robustness of SEU$^2$-Net and its superior performance in liver occupying lesion segmentation, we use SEU$^2$-Net model to train and test on the PUFH and LiTS datasets. Compared to U-Net, AttentionU-Net, UNet3Plus, UNetPlusPlus, U$^2$-Net, DDRNet_23, DNLNet and GSCNN, SEU$^2$-Net achieved superior results in terms of *IoU*, *Acc*, *Kappa* coefficient and *Dice* coefficient (Tables 1 and 2). Compared

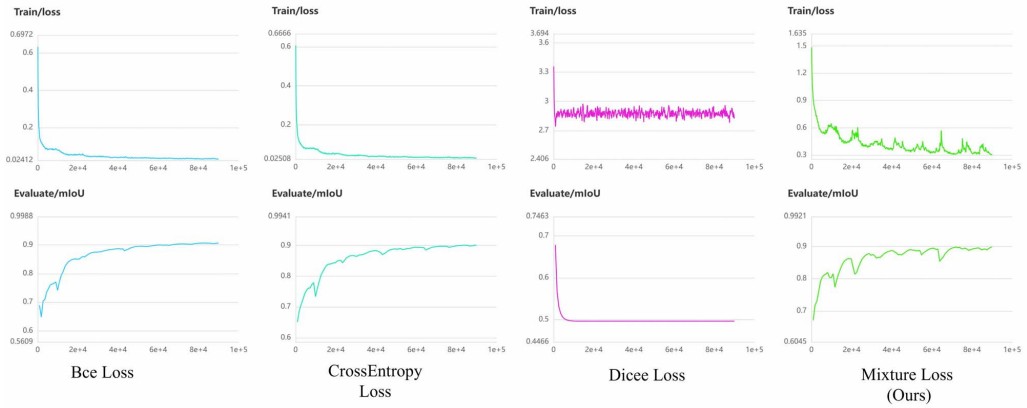

**Figure 8** The loss function and mIoU variation curves for Mixture, CrossEntropy, Dice, and BCE Loss functions on the LiTS dataset.

to other state-of-the-art liver tumor segmentation papers based on attention mechanism deep learning models, *Li et al. (2021)* presented CC-DenseUNet, which incorporated the criss-cross attention (CCA) module into the densely connected U-Net. The CCA attention mechanism gathers contextual information from all pixels along its cross-path, constituting a spatial attention mechanism. On the LiTS dataset, CC-DenseUNet achieved a Dice coefficient of 0.741. *Gong et al. (2022)* proposed a two-tiered U-Net architecture, where two U-Nets are connected through a hard attention module. This module enables the network to automatically crop the input image and remove the corresponding liver region, improving tumor segmentation accuracy. Additionally, spatial and channel attention mechanisms are incorporated into the second UNet. This model achieved a Dice score of 0.798 on the LiTS dataset. *Fan et al. (2020)* introduced the Multi-scale Attention Net (MA-Net), which included the position-wise attention Block (PAB) for spatial attention and the position-wise attention block (PAB) for channel attention. MA-Net achieved a Dice coefficient of 0.749 on LiTS. *Seo et al. (2020)* prepented the modified U-Net (mU-Net), which was composed of object-dependent upsampling, redesigned residual paths, and skip connections to adapt to the segmentation requirements of objects with varying sizes and features. mU-Net attained a Dice coefficient of 0.8972 on the LiTS dataset. In contrast to these four deep learning models based on attention mechanisms, SEU$^2$-Net achieves a Dice coefficient of 0.9093 on LiTS, which indicates that SEU$^2$-Net can more accurately locate and segment liver lesions in the liver occupying lesion segmentation task, while maintaining low error and misclassification rates.

Figures 4 and 6 show that SEU$^2$-Net performs very well in segmenting small and complex liver occupying lesions, which can be attributed to the powerful characteristics of the SE block. The SE block can adaptively adjust the weights of each channel in the feature map to better capture features at different levels. When segmenting small and complex liver lesions, the model requires higher resolution and better feature expression ability, and the introduction of the SE block can effectively improve the feature expression ability of the model, thereby capturing these tiny yet important details. Therefore, SEU$^2$-Net

performs better in the segmentation of small and complex liver occupying lesion. At the same time, by introducing the SE block, the model can better avoid overfitting, improve its generalization ability, and make its performance more stable and reliable on different datasets.

SEU$^2$-Net performs differently on the two datasets. It shows excellent performance on the PUFH dataset, achieving high *IoU*, *Acc*, *Kappa* coefficient, and *Dice* coefficient. However, its performance on the LiTS dataset is relatively lower. This may be due to the differences in characteristics and backgrounds between the two datasets. The PUFH dataset includes various liver occupying lesion such as liver tumors, liver cysts, and liver abscesses, which have relatively obvious shape and texture features, making them easier to capture and distinguish by the model. On the other hand, the liver lesions in the LiTS dataset only include liver tumors, some of which have very small volumes and may be affected by randomness and noise interference, leading to a slight decrease in the model's performance on the LiTS dataset. Overall, SEU$^2$-Net's performance on the LiTS dataset is relatively poor, indicating that there is still room for improvement in SEU$^2$-Net's generalization ability when handling complex datasets.

### Limitations and future work

Although SEU$^2$-Net performs well, there are still some limitations in this study. One key limitation lies in the relatively large number of parameters associated with SEU$^2$-Net when compared to other state-of-the-art methods. This parameter abundance can pose challenges in resource-constrained environments, such as on mobile devices or embedded systems. Future research should focus on optimizing the SEU$^2$-Net architecture to reduce its parameter count while preserving its essential multiscale and attention mechanisms. This optimization aims to make SEU$^2$-Net more adaptable for various medical image segmentation tasks, especially in resource-limited settings. Furthermore, this study's dataset primarily concentrates on liver occupying lesions. Expanding the evaluation of SEU$^2$-Net to encompass larger and more diverse datasets is essential to ensure its generalizability across a broader spectrum of medical image segmentation tasks. Extending the evaluation to diverse datasets will enhance our understanding of SEU$^2$-Net's capabilities and limitations in various medical imaging scenarios.

Future studies should focus on optimizing the SEU$^2$-Net architecture to reduce parameter count while preserving its key features is essential. We plan to study the effects of different SE block configurations and compare them with other attention mechanisms such as CBAM (*Woo et al., 2018*) and BAM (*Park et al., 2018*). Conducting extensive evaluations on diverse datasets will further validate SEU$^2$-Net's applicability in various medical image segmentation scenarios, enhancing its practical utility in the field.

### CONCLUSIONS

We propose a network architecture called SEU$^2$-Net based on improved U$^2$-Net for liver occupying lesion segmentation. SEU$^2$-Net combines the SE block and U$^2$-Net, which retains the ability of U$^2$-Net to capture context information at multiple scales. This model emphasizes useful information and ignores useless information through the

channel attention mechanism. We combine CrossEntropy and Dice Loss functions in the design of our loss function, and test its superiority through ablation and comparative experiments. In addition, we present a new liver occupying lesion CT dataset from Peking University First Hospital's clinical data (PUFH dataset). SEU$^2$-Net is compared with U-Net, AttentionU-Net, UNet3Plus, UNetPlusPlus, U$^2$-Net, DDRNet_23, DNLNet, and GSCNN for liver occupying lesion segmentation on LiTS and PUFH datasets. The *IoU*, *Acc*, *Kappa* coefficient, and *Dice* coefficient of SEU$^2$-Net are 90.55%, 99.70%, 94.89%, 95.04% and 83.37%, 99.86%, 90.86 %, 90.93% on PUFH and LiTS datasets, respectively. The experiment shows that SEU$^2$-Net with attention mechanism outperforms AttentionU-Net with attention mechanism. SEU$^2$-Net can predict small liver occupying lesion and complex contours from the liver occupying lesion segmentation images of the two datasets. Compared with the other networks, the accuracy, repeatability and consistency between the liver occupying lesion segmentation images predicted by SEU$^2$-Net and the label images are the most accurate. Our proposed method would make U$^2$-Net a valuable resource in the field of medical image processing. The superior performance of SEU$^2$-Net with attention mechanism in liver occupying lesion segmentation indicates that it has good development potential in the field of medical image processing.

### Funding

This work was supported by the National Natural Science Foundation of China (Grant No. 61972007) and the Science and technology service network plan of Dongwan Chinese Academy of Sciences (STS) (Grant No. 20211600200102). The funders had no role in study design, data collection and analysis, decision to publish, or preparation of the manuscript.

### Grant Disclosures

The following grant information was disclosed by the authors:
The National Natural Science Foundation of China: 61972007.
Science and technology service network plan of Dongwan Chinese Academy of Sciences (STS): No.20211600200102.

### Competing Interests

The authors declare there are no competing interests.

### Author Contributions

- Lizhuang Liu conceived and designed the experiments, performed the experiments, analyzed the data, performed the computation work, prepared figures and/or tables, authored or reviewed drafts of the article, and approved the final draft.
- Kun Wu conceived and designed the experiments, performed the experiments, analyzed the data, prepared figures and/or tables, authored or reviewed drafts of the article, and approved the final draft.
- Ke Wang conceived and designed the experiments, analyzed the data, authored or reviewed drafts of the article, and approved the final draft.

- Zhenqi Han conceived and designed the experiments, performed the experiments, analyzed the data, performed the computation work, prepared figures and/or tables, authored or reviewed drafts of the article, and approved the final draft.
- Jianxing Qiu conceived and designed the experiments, analyzed the data, authored or reviewed drafts of the article, and approved the final draft.
- Qiao Zhan conceived and designed the experiments, analyzed the data, authored or reviewed drafts of the article, and approved the final draft.
- Tian Wu conceived and designed the experiments, analyzed the data, authored or reviewed drafts of the article, and approved the final draft.
- Jinghang Xu conceived and designed the experiments, analyzed the data, authored or reviewed drafts of the article, and approved the final draft.
- Zheng Zeng conceived and designed the experiments, analyzed the data, authored or reviewed drafts of the article, and approved the final draft.

### Ethics

The following information was supplied relating to ethical approvals (i.e., approving body and any reference numbers):

The Biomedical Research Ethics Committee of Peking University First Hospital approved the study (Ethics Review No. 2020 Scientific Research 101-Amendment)

### Data Availability

The PUFH and LiTS dataset are available at figshare: Liu, Lizhuang; Wu, Kun; Wang, Ke; Han, Zhenqi; Qiu, Jianxin; Zhan, Qiao; et al. (2023). supplementary_files_1.zip. figshare. Dataset. https://doi.org/10.6084/m9.figshare.23312786.v1.

The LiTs dataset is available under CC BY NC-ND 4.0: https://competitions.codalab.org/competitions/17094#learn_the_details-terms_and_conditions.

The LiTS dataset is available at https://competitions.codalab.org/competitions/17094#learn_the_details-overview.

The code is available in the Supplemental File.

### Supplemental Information

Supplemental information for this article can be found online at http://dx.doi.org/10.7717/peerj-cs.1751#supplemental-information.

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
