# Peer review of "SEU2-Net: multi-scale U2-Net with SE attention mechanism for liver occupying lesion CT image segmentation"

_PeerJ Computer Science, doi:10.7717/peerj-cs.1751_

## Round 0.1 · original submission · Major Revisions

The article looks promising but the reviewers noticed a number of issues that need to be addressed. Please take into account their suggested changes and prepare a new version of the manuscript.

Reviewers have requested that you cite specific references. You may add them if you believe they are especially relevant. However, I do not expect you to include these citations, and if you do not include them, this will not influence my decision.

Please note that https://www.hindawi.com/journals/cin/2022/4942637/ has been retracted and should not be cited under any circumstances

**Language Note:** PeerJ staff have identified that the English language needs to be improved. When you prepare your next revision, please either (i) have a colleague who is proficient in English and familiar with the subject matter review your manuscript, or (ii) contact a professional editing service to review your manuscript. PeerJ can provide language editing services - you can contact us at copyediting@peerj.com for pricing (be sure to provide your manuscript number and title). – PeerJ Staff

Reviewer 1 ·

Basic reporting

This paper is written well. Please follow the detailed comments.

Experimental design

Very good.

Validity of the findings

Needs some more statistical results.

Additional comments

The review of the article with the title “SEU2-Net: Multi-Scale U2-Net with SE attention mechanism for liver occupying lesion CT image segmentation”. Liver occupying lesion can have signiûcant consequences for a person9s health and wellbeing. To assist physicians in the diagnosis and treatment planning of abnormal areas in the liver, we propose a novel network named SEU2-Net by introducing the channel attention mechanism into U2-Net for accurate and automatic liver occupying lesion segmentation. We design the SE-RSU block, which is to add the SE attention mechanism at the residual connections of the RSU (the component unit of U2-Net). SEU2-Net not only retains the advantages of U2-Net in capturing context information at multiple scales, but also can adaptively recalibe channel feature responses to emphasize useful feature information according to the channel attention mechanism.
I have the following issues which need a solid response by the authors…
1. Please provide a 3 to 4 points novelty of your work at the end of introduction section.
2. Background work should extend to add recently published deep learning and segmentation papers, for example…
https://doi.org/10.1109/ACCESS.2019.2896961
https://doi.org/10.1109/ACCESS.2021.3056516
https://doi.org/10.1109/ACCESS.2021.3131216
https://doi.org/10.3390/app9010069
https://doi.org/10.1155/2022/7954333
https://doi.org/10.3390/math10050796
https://doi.org/10.1155/2022/4942637
https://doi.org/10.1155/2023/2345835
https://doi.org/10.1007/978-981-10-7389-2_24
https://doi.org/10.1117/12.2540175
https://doi.org/10.1117/12.2540176
3. How the authors selected the best parametric selection, it’s a very hard process to do.
4. Please provide a training and validation graph of the epochs you use during the training network.
5. If possible, please provide a 3D results of liver segmentation.
6. Please extend the discussion section of the paper with other works.

·

Basic reporting
* * *
Experimental design
* * *
Validity of the findings
* * *
Additional comments

Title: SEU2 -Net: Multi-Scale U2 -Net with SE attention mechanism for liver occupying lesion CT image segmentation
This paper proposes a network architecture SEU2 -Net based on improved U2 -Net for liver occupying lesion segmentation. SEU2- Net combines the SE block and U2- Net, which retains the ability of U2-Net to capture context information at multiple scales, and emphasizes useful information and ignores useless information through the channel attention mechanism.
Comments
The contribution of this paper is not clear.
The structure and organization paragraph is missing.
It’s better to add section for the related studies with more details, and add table with some comparison items, year of the publication, methodology, dataset, advantage and disadvantages, and finding results.
Some terms are defined before it used. Please write the full description.
In Figure 1, it’s not clear how the lesions are segmented.
Add more explanation about the attention mechanism used in this study.
It is not clear how authors segment liver in Figure 3 from different positions (coronal, sagittal, axial,…..).
In Figure 3, there is black sub image with no details information.
What about the preprocessing steps?
More numerical analysis is needed and please add statistical significant methods.
More comparisons are needed with stat of the art.
Limitation of this work is missing, in addition, future work section is missing.
The following studies used different segmentation methods for liver and lesion segmentation, please check and refer.
- Anter, A. M., & Abualigah, L. (2023). Deep Federated Machine Learning-Based Optimization Methods for Liver Tumor Diagnosis: A Review. Archives of Computational Methods in Engineering, 30(5), 3359-3378.
- Anter, A. M., Oliva, D., Thakare, A., & Zhang, Z. (2021). AFCM-LSMA: New intelligent model based on Lévy slime mould algorithm and adaptive fuzzy C-means for identification of COVID-19 infection from chest X-ray images. Advanced Engineering Informatics, 49, 101317.
- Anter, A. M., Bhattacharyya, S., & Zhang, Z. (2020). Multi-stage fuzzy swarm intelligence for automatic hepatic lesion segmentation from CT scans. Applied Soft Computing, 96, 106677.
- Elazab, A., Anter, A. M., Bai, H., Hu, Q., Hussain, Z., Ni, D., ... & Lei, B. (2019). An optimized generic cerebral tumor growth modeling framework by coupling biomechanical and diffusive models with treatment effects. Applied Soft Computing, 80, 617-627.
- Anter, A. M., & Hassenian, A. E. (2019). CT liver tumor segmentation hybrid approach using neutrosophic sets, fast fuzzy c-means and adaptive watershed algorithm. Artificial intelligence in medicine, 97, 105-117.
- Anter, A. M., & Hassenian, A. E. (2018). Computational intelligence optimization approach based on particle swarm optimizer and neutrosophic set for abdominal CT liver tumor segmentation. Journal of Computational Science, 25, 376-387.
- Anter, A. M., El Souod, M. A., Azar, A. T., & Hassanien, A. E. (2014). A hybrid approach to diagnosis of hepatic tumors in computed tomography images. International Journal of Rough Sets and Data Analysis (IJRSDA), 1(2), 31-48.
- Anter, A. M., Hassanien, A. E., ElSoud, M. A. A., & Tolba, M. F. (2014). Neutrosophic sets and fuzzy c-means clustering for improving ct liver image segmentation. In Proceedings of the Fifth International Conference on Innovations in Bio-Inspired Computing and Applications IBICA 2014 (pp. 193-203). Springer International Publishing.
- Anter, A. M., ElSoud, M. A., & Hassanien, A. E. (2013, December). Automatic liver Parenchyma segmentation from abdominal CT images. In 2013 9th International Computer Engineering Conference (ICENCO) (pp. 32-36). IEEE.
- Anter, A. M., Azar, A. T., Hassanien, A. E., El-Bendary, N., & ElSoud, M. A. (2013, September). Automatic computer aided segmentation for liver and hepatic lesions using hybrid segmentations techniques. In 2013 Federated Conference on Computer Science and Information Systems (pp. 193-198). IEEE.
About ref [1], is it just one page.

Reviewer 3 ·

Basic reporting

no comment

Experimental design

no comment

Validity of the findings

no comment

Additional comments

This paper introduces a new method for liver segmentation, but the method has only slight improvement on the original method and the experiment and writing are incomplete, so the paper is not suitable for publication in the journal.
1. The only innovation in this article is design the SE-RSU block, which is to add the SE attention at the residual connections of the RSU. This innovation point is very weak for publication.
2. Compared with the trunk model, the effect of the model constructed in this paper is not obvious.
3. Without ablation experiments, it is difficult to say whether the use of both loss functions is effective for segmentation.
4. The words in Figure 4 contain errors.
5. Since U2-Net is a model with a large number of parameters, the article should add some indicators such as parameter number, inference time, FLOPs and so on.
6. The discussion part of the article is only a repetition of the result part, and the content is insufficient.

---

## Round 0.2 · accepted · Accept

The authors address the main points of the reviewers and therefore I can recommend the article to be accepted for publication.

Reviewer 1 ·

Basic reporting

I have no new questions to ask.

Experimental design

The experiments are well understandable.

Validity of the findings

The results are valid.

Additional comments

My all comments have been addressed. I have no new questions to ask.